# Result of Prospective Validation of the Trisomy Test^®^ for the Detection of Chromosomal Trisomies

**DOI:** 10.3390/diagnostics9040138

**Published:** 2019-10-02

**Authors:** Martina Sekelska, Anita Izsakova, Katarina Kubosova, Petra Tilandyova, Erika Csekes, Zaneta Kuchova, Michaela Hyblova, Maria Harsanyova, Marcel Kucharik, Jaroslav Budis, Tomas Szemes, Gabriel Minarik

**Affiliations:** 1Trisomy Test Ltd., Ilkovičova 8, 841 04 Bratislava, Slovakia; anita.izsakova@medirex.sk (A.I.); katarina.kassakova@medirex.sk (K.K.); petra.tilandyova@medirex.sk (P.T.); erikabrezovska@gmail.com (E.C.); zaneta.kuchova@medirex.sk (Z.K.); michalea.hyblova@medirex.sk (M.H.); gabriel.minarik@medirex.sk (G.M.); 2Medirex Inc., Galvaniho 17/C, 821 06 Bratislava, Slovakia; 3Geneton Ltd., Galvaniho 7, 821 06 Bratislava, Slovakia; maria.harsanyova@geneton.sk (M.H.); marcel.kucharik@geneton.sk (M.K.); jaroslav.budis@geneton.sk (J.B.); tomas.szemes@geneton.sk (T.S.)

**Keywords:** NIPT, Trisomy Test^®^, low-coverage whole-genome sequencing

## Abstract

Noninvasive prenatal testing (NIPT) is one of the most common prenatal screening tests used worldwide. Trisomy Test^®^ belongs to NIPT tests based on low-coverage whole-genome sequencing. In our prospective study, 7279 samples of pregnant women collected during approximately two years were analyzed. In this cohort, 117 positive cases for trisomies 21, 18, and 13 were reported. An in-house designed bioinformatic pipeline and proprietary biostatistical approach was used for the detection of trisomies. The pooled sensitivity and specificity of our test reached 99.12% and 99.94%, respectively. The proportion of repeatedly uninformative results after repeated blood draws was 1.11%. Based on the presented results, we can confirm that the Trisomy Test^®^ is fully comparable with other commercial NIPT tests available worldwide.

## 1. Introduction

The prevalence of Down syndrome (caused by trisomy of chromosome 21—T21) is 1 in 800 live births, 1 in 6000 in case of Edwards syndrome (caused by trisomy of chromosome 18—T18) and about 1 in 10,000 newborns suffer from Patau syndrome (caused by trisomy of chromosome 13—T13) [1,2]. Standard scheme for screening for these trisomies in pregnancy is based on biochemical and ultrasound examinations and reaches detection rate between 50% and 95% with a 5% false positive rate [3]. The discovery of the presence of cell-free fetal DNA in maternal plasma in 1997 by Lo et al. [4] and progress in the field of DNA sequencing [5], allowed the development of non-invasive prenatal testing (NIPT) methods that are capable of replacing the standard screening scheme. Different NIPT screening tests use different approaches for the detection of T21, T18, or T13 trisomies. The most commonly used approaches utilize low-coverage whole-genome sequencing (used by, e.g., NIFTY, Verifi) or quantification of fetal-specific single nucleotides polymorphisms (used by, e.g., Panorama, Harmony) analysis. Based on the review of available studies, NIPT reached a sensitivity of >99%, >97%, and 97–99% for detection of trisomy 21, 18, and 13, respectively [6,7]. The pooled specificity for all three trisomies reached 99.9%. The combined false positive rate was 0.13% [7], while the proportion of false negative results was generally very low, at about 0.01% [6].

## 2. Materials and Methods

Peripheral blood samples of pregnant women after the end of 10th week of pregnancy were collected as part of commercially available testing in cooperation with the gynecologist, clinical geneticists, and genetic centers from different countries of Europe and Asia between 1 July 2016 and 30 September 2018. All pregnant women signed informed consent for participation in the prospective validation. In total, 7279 samples were collected in this period. When an uninformative result of the test was reported after analysis of the original blood sample, a second and third blood draw were requested after every 14 calendar days and with respect to the week of pregnancy at the time of the blood draws. For instance, if the first blood draw was performed before the fourteenth week of pregnancy, there was enough time to perform the second, as well as, third blood draw.

Peripheral blood samples were collected into 10 mL EDTA (BD Vacutainer^®^ Plus blood collection tubes, Belliver Industrial Estate, UK, S-Monovette^®^ Sarsted AG&Co. KG, Germany) or STRECK (Cell-Free DNA BCT^®^) tubes and plasma was separated from the blood within 36 h (EDTA) or within 10 days (STRECK) after the blood draw. The EDTA or STRECK tubes were used with respect to sample transportation times, STRECK tubes were used where extended transportation times (longer than 36 h) were expected. Separation of plasma was performed by two-step centrifugation at 2200× *g* and 16,000× *g* (each for 8 min) and the plasma was stored at −20 °C. DNA was extracted using the QIAamp DNA Blood Mini Kit (Qiagen, Germany). The extracted DNA was quantified using a Qubit dsDNA HS Assay Kit (Thermo Fisher Scientific, Oregon, USA). The maximum limit of DNA concentration for samples collected into EDTA tubes was set at 0.3 ng/µL—a higher value was regarded as a result of hemolysis and correlated with a decreased fetal fraction. No limit for DNA concentration was applied in case of DNA extraction from STRECK tubes—no decrease in the fetal fraction was recorded when STRECK tubes were used for blood collection, and blood was processed within 10 days. DNA libraries were prepared using the TruSeq Nano DNA Library Prep Kit (Illumina, San Diego, USA) according to the previously published protocol [8]. A minimum limit of library concentration of 0.1 ng/µL was used in subsequent sequencing. A 2100 Bionalyzer (Agilent, Waldbronn, Germany,) was used for quality control of final libraries. A NextSeq 500/550 platform and High Output Sequencing Kit v2 (75 cycles) (Illumina) with pair-end sequencing protocol (2 × 35 cycles) were used for sequencing.

The first parts of sequencing data processing (included mapping, correction of GC bias, and fetal fraction calculation) were performed as published previously [8]. Next, z-scores for target chromosomes 21, 18, and 13 were calculated from the resulting chromosomal vectors as previously published [9]. Aneuploidies or aberrations of other chromosomes (e.g., sex chromosomes, 22q11.2 deletion) were not evaluated in this study. Samples with a z-score higher than 4.0 were reported as positive for chromosome aneuploidy. Samples with all z-scores below 2.5 were reported as negative. A sample was evaluated as uninformative in cases of z-score between 2.5 and 4.0 (assay failure) or with fetal fraction below 5%.

## 3. Results

In this prospective study, 7279 samples of pregnant women with singleton and twin pregnancies were analyzed. Gestation age was between 10 + 0 and 33 + 0 weeks of pregnancy. For all samples, z-scores for targeted chromosomes (21, 18, and 13) were calculated. Samples with z-score below 2.5 were reported as low-risk (negative), those with z-score between 2.5 and 4 were reported as uninformative (assay failure), and those with z-score above 4 were reported as high-risk (positive) for the corresponding chromosome trisomy. The structure of informative and uninformative results is summarized in Figure 1.

T21 was the most frequently detected trisomy reported in 84 cases. Less frequently reported trisomies included were T18 (21 cases) and T13 (12 cases). Out of 117 positive samples, four were not confirmed afterward, and were, therefore, classified as false positives (1 case of T21 and 3 cases of T13). The overall false positive rate (FPR) for all three trisomies was 0.06%. The overall specificity of the method reached 99.94%. One case with false negative result for T21 was recorded in the cohort and the false negative ratio (FNR) thus reached 0.88%. The overall sensitivity of the method was 99.12%. Statistical evaluation of trisomy detection is summarized in detail in Table 1.

## 4. Discussion

7279 pregnant plasma samples analyzed between 1 July 2016 and 30 September 2018 were included in our prospective study. All samples were analyzed using the same laboratory protocol and sequencing platform to obtain data with the same technical characteristics. Identical analytical settings were used for both types of blood collection tubes (EDTA or STRECK). Although three different types of Trisomy test were included in the study (Trisomy test focused on T21, T18, and T13 only; Trisomy test XY that additionally focuses on aneuploidies of sex chromosomes; and Trisomy test +, which furthermore tests for five microdeletion syndromes), we only evaluated detection statistics for the three common trisomies—T21, T18, and T13. The reason for not evaluating the detection accuracy of sex chromosome aneuploidies as well as other microdeletions in this study was that the screening for them started later and lacked any positive samples. Thus sensitivity, specificity, positive predictive value (PPV), and negative predictive value (NPV) calculations were not statistically relevant. Nevertheless, the pooling of results from these three types of tests should not be associated with biased data regarding the targeted trisomies. However, it may lead to bias in the estimation of the proportion of uninformative results because the pool of uninformative samples also contained uninformative results regarding sex chromosome aneuploidies and microdeletions. Given the facts, the evaluation of uninformative results is actually overestimated when it comes to common trisomies.

In our cohort, sensitivity for T21 was 98.81% (95% Cl, 93.5–99.9%), which was very similar to results from the recently published meta-analyses—99.7% (95% CI, 99.1–99.9%) and 99.8% (95% CI, 98.1–99.9%) [6,7]. The specificity of T21 detection was 99.99% in our case (95% CI, 99.9–100%) and very similar for both meta-analyses 99.9% (95% CI, 99–99.9%) [6,7]. Due to the lack of positive cases for T18, differences in sensitivity between available meta-analyses were small—97.9% (95% CI, 94.9–99.1%) and 97.7% (95% CI, 95.8–98.7%) [6,7], while our results reported 100% (95% Cl, 61.9–100%). In the meta-analyses sensitivity for the least frequent T13 was 99.0% (95% CI, 65.8–100%) and 97.5% (95% CI 81.9–99.7%) [6,7], corresponding to 100% (95% Cl, 45.7–100%) in our cohort. Specificity of both T18 and T13 was reported at 99.9% for both meta-analyses and 100% and 99.96% in our cohort, respectively. As T18 and T13 cases are not as common, the 100% sensitivity of our test could be biased. This limitation of statistical evaluation of sensitivity and specificity for T18 and T13 was also mentioned in both meta-analyses [6,7]. The overall sensitivity for all three trisomies in our study was 99.12% (95% Cl, 81.7–100%), which was comparable to 99.9% (95% CI, 99.8–99.9%) in the meta-analysis [6]. The overall false positive rate in our study was reported at 0.06%, which was comparable with 0.1% and 0.13% published in the meta-analyses [6,7]. The reason for false positivity (1 case of T21 and 3 cases of T13) and false negativity (1 case of T21) remained mostly unknown as the placental samples for one T21 case and two cases of T13 were not available for further testing for potential fetoplacental mosaicism. In the third T13 case, a marker chromosome was detected using cytogenetic confirmatory analysis after amniocentesis. No clinical follow-up nor confirmatory testing after delivery were obtained in the only false negative case.

The proportion of uninformative results was 4.7% after the first blood draw, which is in concordance with the meta-analyses, varying from 0.9% to 4.6% and from 0% to 12.2% [6,7]. The most common cause of the uninformative results in our study was a low fetal fraction (67.9%). In 28.6% of these cases, it was due to high maternal weight (≥90 kg) (average weight of pregnant women in our cohort was 67.4 kg). It is known that gestational age and maternal weight correlates with fetal fraction and that repeated blood draw could be used to overcome this problem in a significant portion of such cases [10]. The blood samples taken after 2 or 4 weeks following the original blood draw became informative in 73.5% of cases, which was more than the 60% reported in the meta-analysis [7]. The final failure rate of the Trisomy test^®^ (1.11%) was fully comparable with the published studies, where the failure rate ranged from 0.1% to 8.1% with the median at 2.4% [7].

Based on the presented data, we conclude that the performance of the Trisomy test^®^ with focus on sensitivity, specificity, and the failure rate is comparable to other available tests used in NIPT worldwide. It could provide useful insight with potential implications for prenatal screening for syndromes, for many of which there is currently no other screening option available.

## Figures and Tables

**Figure 1 diagnostics-09-00138-f001:**
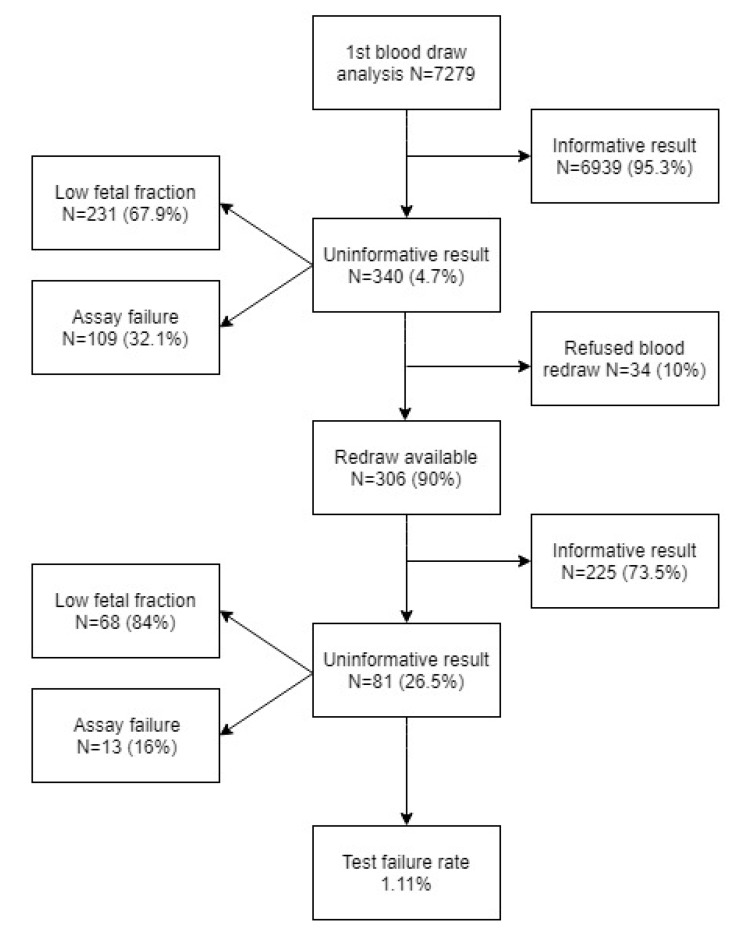
Structure of sample cohort regarding informative and uninformative results.

**Table 1 diagnostics-09-00138-t001:** Statistics of detection of chromosome 21, 18, and 13 trisomies. PPV—positive predictive value, NPV—negative predictive value, FPR—false positive rate.

Result	Trisomy 21	Trisomy 18	Trisomy 13	Overall Results
True positive	83	21	9	113
False positive	1	0	3	4
True negative	6854	6918	6927	6821
False negative	1	0	0	1
Sensitivity	98.81%	100.00%	100.00%	99.12%
Specificity	99.99%	100.00%	99.96%	99.94%
PPV	98.81%	100.00%	75.00%	96.58%
NPV	99.99%	100.00%	100.00%	99.99%

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
