# Peer review of "Result of Prospective Validation of the Trisomy Test® for the Detection of Chromosomal Trisomies"

_diagnostics, 2019, doi:10.3390/diagnostics9040138_

Round 1

Reviewer 1 Report

While the results of this study are worthwhile, there is an overriding need for the paper to be reviewed by a person with excellent skills in English, particularly in sentence structure and content.  There are too many sentences that must be re-structured, particularly since there are sentences with incorrect word-choice or that lack predicates.

Author Response

While the results of this study are worthwhile, there is an overriding need for the paper to be reviewed by a person with excellent skills in English, particularly in sentence structure and content.  There are too many sentences that must be re-structured, particularly since there are sentences with incorrect word-choice or that lack predicates.

Re: Manuscript was reviewed by person skilled in English and many sentences corrected and restructured according to the reviewer´s recommendations.

Reviewer 2 Report

In this prospective study Martina Sekelsa et al report the data of the analysis of 7279 samples of pregnant women using their test Trisomy Test®. Their results suggest that this test is comparable with others in the market.

Comments:

Results:

In this section the first data that I would describe would be the percentage of informativity of the test (also the number of informative and non-informative samples). Then I would follow with the ploidy results. Also, I think is important to mention in the results that this test allows only the analysis of 3 chromosomes, 21, 18 and 15. What about sex chromosomes? The last paragraph of this section, specially when authors explain that some women required 2 and 3 extra blood draws is difficult to follow. Please consider rewriting.

Material and methods:

Regarding the type of tube used (STRECK or EDTA), why do authors use two types of tubes? Have authors test that there are not differences in the results using one or the other tube? Also, samples are left in the STRECK tubes 10 days. In our hands, 3 days is the maximum. Have authors performed time tests? The DNA concentration when EDTA tubes are used is 0.3ng/ul. For STRECK tubes? Please, If the same, please mention it, if not, explain why. Also, statically studies should be performed to check that indeed there are no differences in all the values among chromosomes and techniques, as mentioned in discussion (line 72).

Discussion:

If samples have been performed in different laboratories it should be stated, and data divided per laboratory. In lines 73-80 authors said that microdeletions and sexual chromosomes has been analyzed. Why the data is not shown? If authors want to state this in the discussion this data should appear at least in results. As authors claim that ICM is important in the uninformative results, the average weight of the women should be stated in material and methods. In the last paragraph authors concluded that this “currently used sample processing protocol as well as bioinformatic analysis enable its utilization in NIP Screening focused NOT only on reported most commonly trisomies”. This is misleading, since no data regarding other chromosomes (but 12, 18 and 21) appear in the paper.

Minor comments:

There are several misspelling and grammar mistakes, please if authors can correct the English would be perfect.

Author Response

Results:

In this section the first data that I would describe would be the percentage of informativity of the test (also the number of informative and non-informative samples). Then I would follow with the ploidy results.

Re: Result section was updated and “informativity of the test” part was moved to the beginning of the section as recommended.

Also, I think is important to mention in the results that this test allows only the analysis of 3 chromosomes, 21, 18 and 15. What about sex chromosomes?

Re: Corresponding information related to estimation of statistical parameters of only the three aneuploidies was added to Discussion section. Of course, the test in currently used version is reports also sex chromosomes aneuploidies, but as we have started with sex chromosome evaluation later and the number of samples with reported and confirmed/not confirmed detections is low it is not possible to perform evaluation on it. It should be done later on, when the number will reach statistically significant level. Same situation is in the panel of selected microdeletions detection, the evaluation I ongoing and reasonable number could be reached at the end of 2020.

The last paragraph of this section, specially when authors explain that some women required 2 and 3 extra blood draws is difficult to follow. Please consider rewriting.

Re: Texts related to the redraws was rewritten, we hope that now it is not only more detailed but also clearly described in Materials and Methods section.

Material and methods:

Regarding the type of tube used (STRECK or EDTA), why do authors use two types of tubes?

Re: Different tube types were used in relation to the sample to lab transportation times. In cases with transportation time allowed up to 36 hours delivery of the sample to the lab blood draw was managed using EDTA tubes, when it should be longer, STRECK tubes were used.

Have authors test that there are not differences in the results using one or the other tube?

Re: We have performed confirmatory parallel sampling tests that showed that there is no difference in analyses of samples collected to EDTA and STRECK tubes at the beginning of usage of STRECK tubes. Both fetal fraction and sensitivity and specificity of detections were compared and no significant differences were recorded.

Also, samples are left in the STRECK tubes 10 days. In our hands, 3 days is the maximum. Have authors performed time tests?

Re: Yes, we have performed tests comparing different times of sample storage in STRECK tubes and there were no significant differences in fetal fraction as well as detections for up to 10 days, therefore this limit is applied in out blood samples. Our results are in concordance with the original data presented by tube manufacturer, that showed 14 days stability of the samples at room temperature.

The DNA concentration when EDTA tubes are used is 0.3ng/ul. For STRECK tubes? Please, If the same, please mention it, if not, explain why. Also, statically studies should be performed to check that indeed there are no differences in all the values among chromosomes and techniques, as mentioned in discussion (line 72).

Re: Before utilizing of Trisomy test and EDTA or STRECK tubes in routine testing the limit for EDTA tubes originated DNA extracted samples at 0.3 ng/ul was confirmed as limit for identification of samples with low fetal fraction caused by hemolysis of blood samples. No similar limit was observed when STRECT tubes were used. Therefore, the limit is only used as QC step for EDTA originated DNA samples. Corresponding information was added to the Materials and Methods section.

Discussion:

If samples have been performed in different laboratories it should be stated, and data divided per laboratory.

Re: All samples were analyzed in one laboratory, with the same laboratory protocol and bioinformatical pipeline.

In lines 73-80 authors said that microdeletions and sexual chromosomes has been analyzed. Why the data is not shown? If authors want to state this in the discussion this data should appear at least in results.

Re: The corresponding information was added to the Discussion section, the reason for evaluating only the three trisomies, although XY aneuploidies and microdeletions were analyzed is, that the screening for XY aneuploidies and microdeletions started later (during the monitored time frame) and so results for evaluation these other aberrations are lacking reasonable counts of positive samples enabling relevant statistical analysis. Prospective study focused on XY aneuploidies as well as microdeletions detection is still ongoing and results could be evaluated at the end of 2020.

As authors claim that ICM is important in the uninformative results, the average weight of the women should be stated in material and methods.

Re: The corresponding information about average weight of the women was added to corresponding sentence in Discussion section, so readers of the paper will be confronted with it directly in the part, where uninformative cases related to maternal weight are discussed.

In the last paragraph authors concluded that this “currently used sample processing protocol as well as bioinformatic analysis enable its utilization in NIP Screening focused NOT only on reported most commonly trisomies”. This is misleading, since no data regarding other chromosomes (but 13, 18 and 21) appear in the paper.

Re: The “conclusion” paragraph was rewritten and now should summarize the results of the study as well as mention possible future development in the field of NIPT focused on other chromosomal aberrations.

Minor comments:

There are several misspelling and grammar mistakes, please if authors can correct the English would be perfect.

Re: Manuscript was reviewed by person skilled in English and many sentences corrected and restructured according to the reviewer´s recommendations.

Round 2

Reviewer 1 Report

Article acceptable in present form.

Author Response

Manuscript was sent to professional proofreading service, so hopefully there are no more errors or words misspelled remaining
